# Prediction of True Stress at Hot Deformation of High Manganese Steel by Artificial Neural Network Modeling

**DOI:** 10.3390/ma16031083

**Published:** 2023-01-26

**Authors:** Alexander Yu. Churyumov, Alena A. Kazakova

**Affiliations:** Department of Physical Metallurgy of Non-Ferrous Metals, National University of Science and Technology “MISiS”, Leninskiy Prospekt 4, 119049 Moscow, Russia

**Keywords:** artificial neural network, hot deformation, thermomechanical simulator Gleeble, high Mn steel, constitutive model

## Abstract

The development of new lightweight materials is required for the automotive industry to reduce the impact of carbon dioxide emissions on the environment. The lightweight, high-manganese steels are the prospective alloys for this purpose. Hot deformation is one of the stages of the production of steel. Hot deformation behavior is mainly determined by chemical composition and thermomechanical parameters. In the paper, an artificial neural network (ANN) model with high accuracy was constructed to describe the high Mn steel deformation behavior in dependence on the concentration of the alloying elements (C, Mn, Si, and Al), the deformation temperature, the strain rate, and the strain. The approval compression tests of the Fe–28Mn–8Al–1C were made at temperatures of 900–1150 °C and strain rates of 0.1–10 s^−1^ with an application of the Gleeble 3800 thermomechanical simulator. The ANN-based model showed high accuracy, and the low average relative error of calculation for both training (5.4%) and verification (7.5%) datasets supports the high accuracy of the built model. The hot deformation effective activation energy values for predicted (401 ± 5 kJ/mol) and experimental data (385 ± 22 kJ/mol) are in satisfactory accordance, which allows applying the model for the hot deformation analysis of the high-Mn steels with different concentrations of the main alloying elements.

## 1. Introduction

High-manganese lightweight steels are prospective materials for the automotive industry due to their elevated specific strength and low cost. Such materials possess high strength, plasticity, and toughness at room temperature [1,2,3,4]. Moreover, they show high energy absorption during impact loads, which may have a significant effect during automobile traffic accidents [5]. The properties of the steel are determined by the chemical composition and producing technology, including high-temperature deformation [6,7]. Therefore, the hot deformation behavior of the alloys requires a deeper investigation.

Hot deformation significantly changed the initial microstructure of the cast ingots by eliminating the casting defects and refining grains by dynamic recrystallization [8]. However, the optimization of the hot deformation conditions is necessary to obtain the required microstructure and subsequently good mechanical properties. The constitutive modeling of the high-temperature behavior may decrease the time for process optimization. They allow for the determination of the rheological properties of the materials at any thermomechanical parameter, such as temperature, strain, and strain rate. Such models may also be useful for the simulation of the deformation by the finite element method [9,10,11,12,13,14]. Currently, a lot of constitutive models have been constructed for metallic materials [15,16,17,18]. Yang et al. have found the dependence of true stress on the Zener–Hollomon parameter during the Fe–27Mn–11.5Al–0.95C steel hot deformation [19]. Wan et al. have established a physical constitutive model considering strain coupling for Fe–25Mn–10Al–1.46C [20]. Shen et al. have investigated high-temperature tensile behavior of as-cast high-Mn steels through constitutive modeling using the Zener–Hollomon parameter [21]. However, most of the models have considered materials with specific chemical compositions. A more common problem in physical metallurgy is finding the influence on the behavior at hot deformation, with both effects of thermomechanical parameters and chemical composition simultaneously. Unfortunately, there are no mathematical models that describe the influence of element concentration on the stress at high temperatures due to the complexity of the deformation at high temperatures. The application of machine learning through artificial neural network (ANN)-based models may help to solve this task.

ANN modeling provides a powerful instrument for finding correlations between properties and influencing parameters without preliminary stated functional dependencies. Due to a lot of non-linear connections between neurons, ANN has significantly higher accuracy in comparison with usual regression models. ANN-based models have found a wide application in metallic materials science for the last time [22,23,24,25,26,27,28,29]. U. Subedi et al. have determined the presence of an intermetallic phase in multi-principal element alloys by ANN modeling [30]. X. Geng et al. have predicted the hardenability of non-boron steels by machine learning [31]. W. Choi et al. have used the ANN approach to predict the influence of vanadium content on the microstructure and mechanical properties of low-alloyed high-strength steel [32]. P. Opela et al. have applied the deep learning of an ANN-based model to describe the hot flow stress of 38MnVS6 steel [33]. Jeong et al. have constructed a model for the prediction of the hot ductility region in high-Mn steel [34]. Cheng et al. have shown significantly higher accuracy of the ANN-based model of the GH4169 superalloy’s warm deformation behavior in comparison with the Arrhenius equation [35]. An analogous result was found by Liu et al. for 42CrMo steel [36]. However, most of the authors have not used the full power of the ANN-based modeling using this approach for the steels with a specific composition.

Therefore, this study aims to construct an ANN-based model for the prediction of high Mn lightweight steel high-temperature deformation behavior with different alloying elements content. Such a model will be useful for the creation of more effective hot deformation technologies for industry.

## 2. Materials and Methods

The ANN model was constructed using the data from scientific papers devoted to the hot deformation behavior of high-Mn light-weight steels [19,20,37,38,39,40,41,42,43,44,45,46,47,48,49,50,51,52]. The database consists of the values of input variables: alloying element content (C, Si, Mn, Al) and thermomechanical parameters (strain rate, temperature, strain), and an output property (true stress). The organizational structure of the constructed ANN model is shown in Figure 1. The ranges of the input parameters in the database are given in Table 1. The number of records in the database was 3648. The obtained data were mixed randomly and separated into the following groups: the training data (60% of the dataset), the cross-validation records (20%), and the data for testing (20%). The transfer function in the neurons was the hyperbolic tangent. The static backpropagation algorithm was used for determining the optimal ANN-based model weight values using NeuroSolutions 7 software.

Validation of the constructed model was conducted by carrying out the additional hot deformation test with the Fe–28Mn–8Al–1C (wt. %) steel. The samples for the deformation with a radius of 3 mm and a height of 9 mm were made from the ingots, which were produced using commercial purity raw materials by argon induction melting in an Indutherm 20 V furnace. The compression was carried out using a Gleeble 3800 thermomechanical simulator at temperatures of 900–1150 °C and strain rates of 0.1, 1, and 10 s^−1^. The final strain was about 1. The thermomechanical treatment process in more detail is described in [44]. The obtained primary stress-strain curves were corrected for considering the friction between dies and the sample’s edges, such as adiabatic heating during the compression, accordingly [53,54].

A Tescan-VEGA3LMH scanning electron microscope (SEM) and Bruker Advance D8 X-ray diffractometer were used for the microstructural analysis. The samples for SEM were polished and chemically etched in the 5% nitric acid solution in alcohol. The mass fraction of phases at high temperatures was determined using the Thermocalc^®^ program with the TCFe7 thermodynamic database.

## 3. Results

### 3.1. Training of the ANN-Based Model

The primary database was used for the estimation of the precision of the constructed model. Figure 2 shows the plots of the calculated and real stress values for training, cross-validation, and testing datasets. The quantitative precision of the constructed model was found using average relative error (ARE) [55]:
(1)$$ARE\left(\%\right)=\frac{100}{N}\overset{N}{\underset{i=1}{\sum }}\frac{\left|E_{i}-P_{i}\right|}{E_{i}}$$

Here, *E*, and *P* are the real and predicted stress, correspondently. *N* is the number of records in the dataset.

The ARE has a value of 5.4% for the training dataset and 6.3% for the cross-validation and testing datasets. Such a low error shows the good accuracy of the built model. However, the error values were obtained for the data that was used for the ANN teaching. It may be useful to check the constructed ANN-based model using an independent experiment for the steel with a composition that differs from the one presented in the database.

### 3.2. Microstructure of the Investigated Steel

New hot compression tests of the Fe–28Mn–8Al–1C steel were carried out to verify the accuracy of the constructed ANN model. The microstructure of the steel before deformation is shown in Figure 3. The microstructure is represented by grains with a size of 41 ± 4 μm (Figure 3a). The only austenite phase was determined by XRD analysis (Figure 3b). Thermodynamic modeling of the phase composition shows that the microstructure of the considered steel in the temperature range of deformation consists of the austenite (fcc) phase (Figure 4).

### 3.3. High-Temperature Deformation Behavior

The Fe–28Mn–8Al–1C steel shows typical metallic materials’ hot deformation behavior (Figure 5). True stress tends to increase with a decrease in temperature and an increase in the deformation rate. The movement of the dislocation at high temperatures is determined by diffusion and applied forces. As a result, a decrease in the temperature leads to the necessity of applying higher stress for the deformation continuation. At the same time, each elemental step of the dislocation movement (sliding or climbing) requires activation time. The increases in the strain rate make the time shorter; as a result, activation of the dislocation movement may proceed only at a higher strain rate. Moreover, the peak is present on all of the true stress–true strain dependences. It means that the process of dynamic recrystallization (DRX) begins during the deformation. However, the DRX process proceeds in the overall volume of the deformed samples only at high temperatures. In that case, we can see steady-state deformation with an almost constant true stress value.

### 3.4. Approvement of the Constructed ANN-Based Model

The constructed ANN model was used for the calculation of the true stress for the additional compression tests of the Fe–28Mn–8Al–1C steel. A comparison of the experimental and predicted true stress values is shown in Figure 6. The ANN model has shown satisfactory accuracy with an ARE of 7.5% (Figure 6d). The highest discordance between experimental and calculated values was observed for a large strain value. It may be related to the significant influence of friction at high strain that is hard to predict correctly in all cases. However, usually, the deformation amount for each elemental stage in the industrial processes does not exceed the value of engineering deformation of about 50% (0.7 of true strain). Up to this value, the model shows good accuracy for almost all investigation deformation modes.

## 4. Discussion

A constructed ANN-based model may be used for the prediction of the flow stress at different compression conditions for new high-Mn steel with a composition that differs from that investigated earlier. As shown in Figure 7a, the true stress of the Fe–xMn–8Al–1C steel increased with increases in Mn content from 20 to 30%. However, similar dependence for the steel with variable Al content has a maximum of about 8% (Figure 7b). It may be related to the difference in the phase compound of the considered alloys. In the case of the Fe–(20-30)Mn–8Al–1C steel, the only austenite phase is present in all temperature and concentration ranges (Figure 7c). The main factors that determine the increase in true stress are solid solution hardening and an increase in stacking fault energy (SFE). The increase in Mn content leads to increases in the SFE [56] and suppresses the DRX process, which is the main softening mechanism at high deformation. The addition of Al to high-Mn steels also significantly increases the SFE of austenite [57]. However, the phase composition of the steel is changing at a concentration of Al of about 8%. The appearance of the ferrite phase leads to a decrease in true stress. Similar hot deformation behavior is observed with the change in carbon concentration (Figure 8a,c). The decrease in carbon content leads to an increase in flow stress due to the high diffusivity of the carbon. However, the appearance of the ferrite in the microstructure leads to a decrease in flow stress. The addition of Si to the investigated alloy drastically decreases flow stress. The Si has also large diffusive mobility in the austenite phase, which leads to the acceleration of softening processes, such as dynamic recovery and dynamic recrystallization.

The built ANN-based model also may help to analyze the deformation behavior of the steel through the effective activation energy (EAE) of the deformation process. The value of the EAE (*Q*) is usually determined using the functional dependence between the stress (σ) and the Zener–Hollomon parameter (*Z*) [58]:(2)$$Z=\dot{\varepsilon }e^{\frac{Q}{RT}}$$
where $\dot{\varepsilon }$, and *T* are deformation rate (s^−1^) and temperature (K), correspondently. Usually, the hyperbolic sine law may describe the relation (2) at all stress values:(3)$$Z=A_{3}{\left[\sinh\left(\alpha \sigma \right)\right]}^{n_{2}}$$
where $A_{3}$*,* $n_{2}$ and α are the material’s constants. However, the description of particular cases of the deformation modes is necessary to define the *α* value. The exponential law may be used for high stresses:(4)$$Z=A_{2}e^{\beta \sigma }$$

The power form well defines the deformation conditions with a low level of stress:(5)$$Z=A_{1}\sigma^{n_{1}}$$
where material’s constants $A_{1}$, $n_{1},$ $A_{2},$ and β should be determined using the values of the flow stress.

The value of α may be found using the following formula:(6)$$\alpha \approx \frac{\beta }{n_{P}}$$

The value of the EAE was determined by logarithmization of Equations (2)–(5) and minimization between predicted and real true stress values by the least squares method.

The experimental value of EAE for Fe–28Mn–8Al–1C steel is 385 ± 22 kJ·mol^−1^. The dependence of the EAE on the Mn concentration in the Fe–xMn–8Al–1C steel was obtained using calculated ANN-based model data. As shown in Figure 9a, the difference between the calculated and experimental values of EAE is within the confidence interval. A good accordance between experimental and calculated data was also shown for the Fe–26Mn–8Al–lC steel [49]. The calculated dependence shows that the deformation process proceeds with more difficulty with the increased Mn concentration. The change in the Mn concentration has also had a significant influence on the other parameters of the constitutive model (Appendix A). The increase in Mn content leads to an increase in all parameters. However, the constants related to strain rate sensitivity (n_1_, n_2_, α, β) have a maximum in the range of the Mn concentration of 24–28%. The same position of the maximum was obtained in the dependence of the activation volume ($3\sqrt{3}kT\left(\frac{\partial \left(ln\dot{\varepsilon }\right)}{\partial \sigma }\right)$ [59] on the Mn content at different temperatures (Figure 9b). The nature of this phenomenon requires detailed study.

Thus, a constructed ANN model for the prediction of the true stress of lightweight high-Mn steel may be useful for the creation of optimal forming technologies and the scientific analysis of the hot deformation behavior.

## 5. Conclusions

An ANN model for the prediction of the hot deformation behavior of the lightweight high-Mn steel was built. The model possesses high accuracy for the training, cross-validation, and testing datasets. An error of prediction in the range of 5.4–6.3% shows the high accuracy of the built model.The additional compression tests of the Fe–28Mn–8Al–1C steel were made for verification of the constructed ANN model. The matching of the calculated and experimental values shows a high model predictability at a true strain of up to 0.7.The effective activation energies for calculated and experimental true stress data for a strain of 0.7 were determined using the dependence between stress and the Zener–Hollomon factor. The effective activation energy values for predicted (401 ± 5 kJ·mol^–1^) and experimental data (385 ± 22 kJ·mol^−1^) are in satisfactory accordance, which allows applying the model for the high-temperature compression behavior analysis of the high-Mn steels with different concentrations of the main alloying elements.The usage of the constructed model shows that the increases in Mn in the Fe–xMn–8Al–1C steel from 20 to 30% lead to increases in the true strain at a deformation rate of 0.1 s^−1^ and true strain of 0.7. Similar dependence for the Fe–28Mn–(5-10)Al–1C has a maximum at the Al content of 8% due to a change in the phase composition from the austenite to the austenite–ferrite region.The dependence of the activation volume on the manganese content has a maximum near the 26% of Mn that may be related to the SFE dependence change and is required following investigation.

## Figures and Tables

**Figure 1 materials-16-01083-f001:**
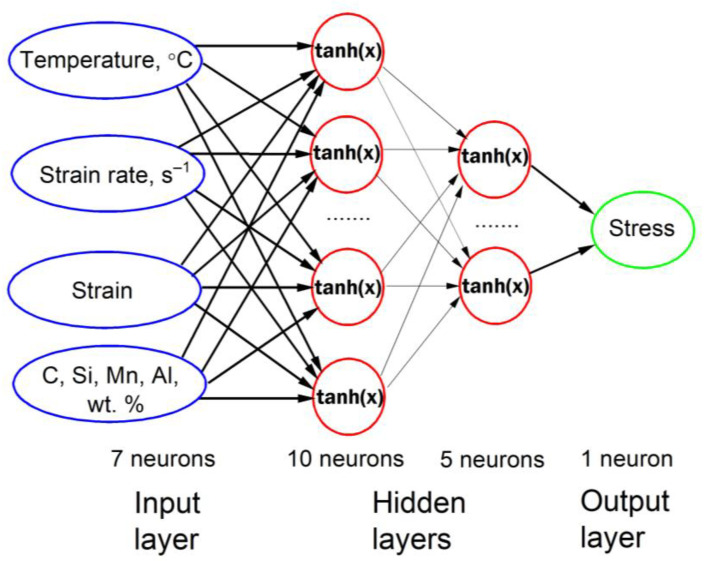
The ANN model organization structure.

**Figure 2 materials-16-01083-f002:**
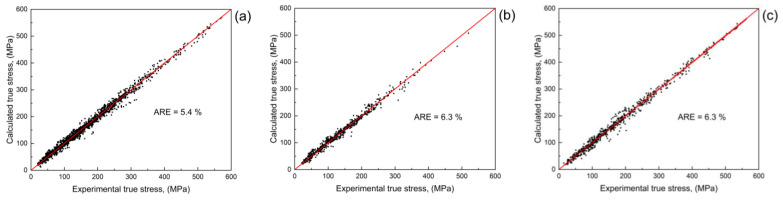
A matching between calculated using a built ANN-based model and real true stresses for (**a**) training dataset, (**b**) cross-validation dataset, and (**c**) testing dataset.

**Figure 3 materials-16-01083-f003:**
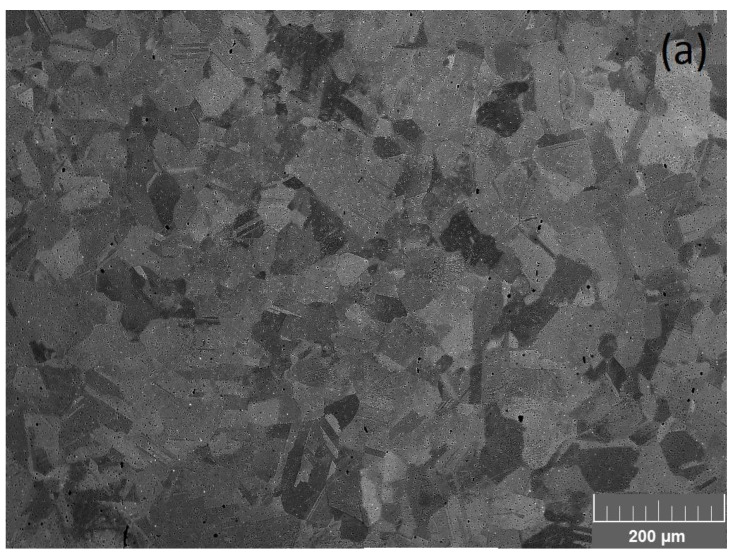
Grain microstructure and phase composition of the Fe–28Mn–8Al–1C steel before deformation: SEM (**a**) and XRD pattern (**b**).

**Figure 4 materials-16-01083-f004:**
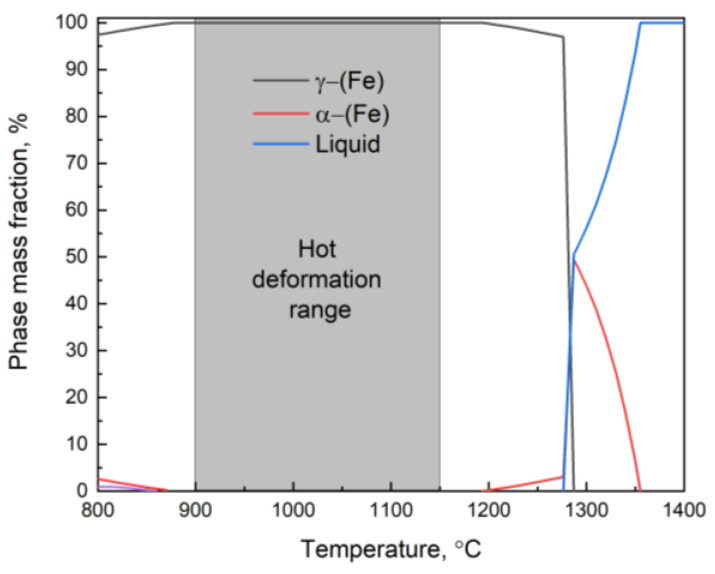
The calculated phase composition at high temperatures of the investigated steel.

**Figure 5 materials-16-01083-f005:**
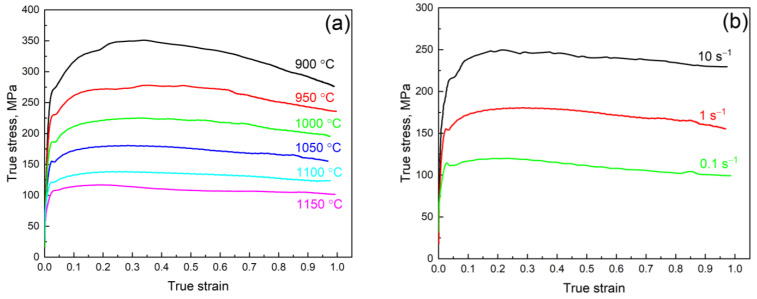
Typical true stress–true strain curves for the Fe–28Mn–8Al–1C steel at a temperature of 1050 °C (**a**) and a deformation rate of 1 s^−1^ (**b**).

**Figure 6 materials-16-01083-f006:**
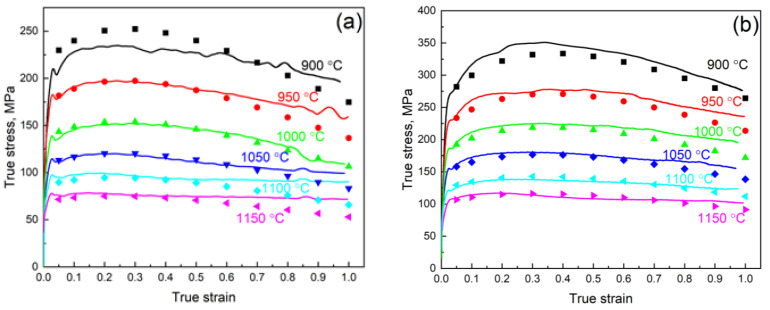
A matching of the experimental (lines) and calculated (dots) stress for the Fe–28Mn–8Al–1C steel deformation at a deformation rate of 0.1 s^−1^ (**a**), 1 s^−1^ (**b**), 10 s^−1^ (**c**), and for all thermomechanical modes (**d**).

**Figure 7 materials-16-01083-f007:**
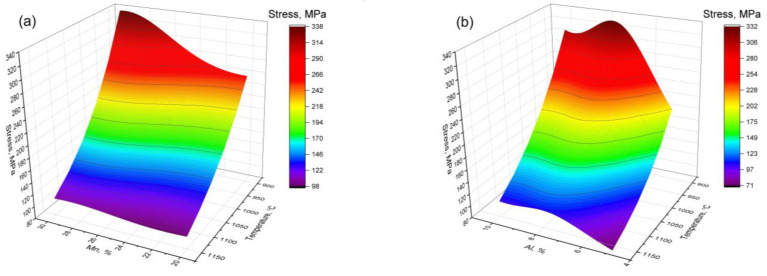
Calculated dependence of the true stress on the temperature and Mn (**a**) and Al (**b**) contents at a strain rate of 1 s^−1^ and a true strain of 0.7. The phase diagrams of the Fe–xMn–8Al–1C (**c**) and Fe–28Mn–xAl–1C (**d**).

**Figure 8 materials-16-01083-f008:**
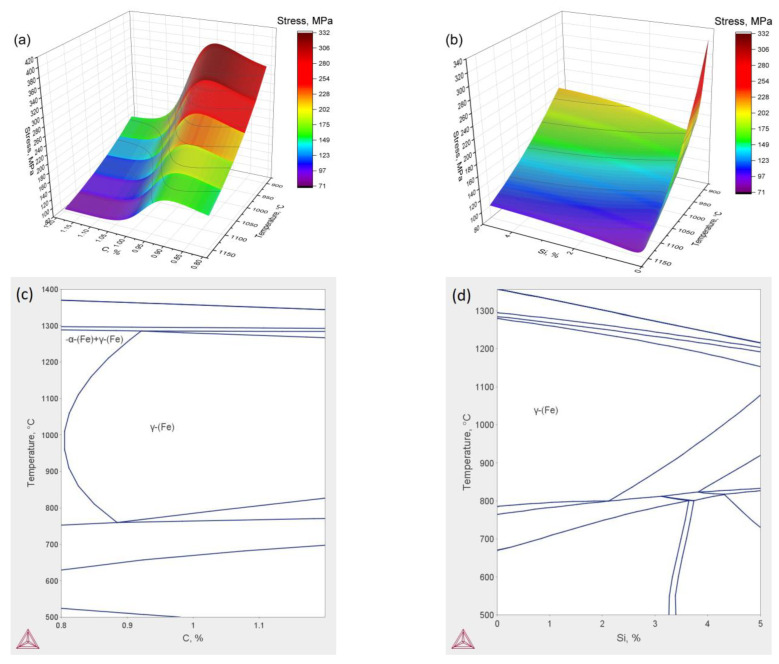
Calculated dependence of the true stress on the temperature and C (**a**) and Si (**b**) content at a strain rate of 1 s^−1^ and a true strain of 0.7. The phase diagrams of the Fe–28Mn–8Al–xC (**c**) and Fe–28Mn–8Al–1C–xSi (**d**).

**Figure 9 materials-16-01083-f009:**
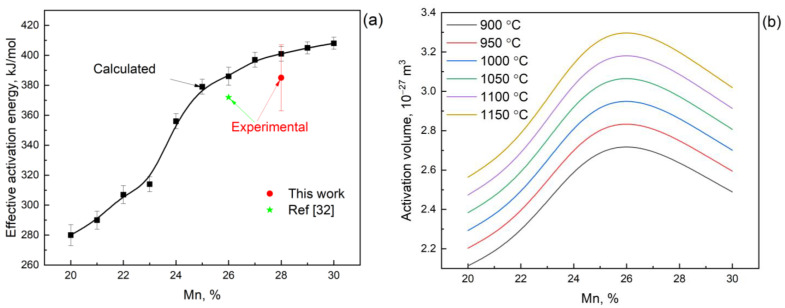
Dependence of the EAE (**a**) and activation volume (**b**) on the Mn content in the Fe–xMn–8Al–1C steel at a true strain of 0.7.

**Table 1 materials-16-01083-t001:** The ranges of the C, Si, Mn, and Al content (wt. %) and hot deformation parameters in the database.

C	Si	Al	Mn	Strain	Temperature, °C	Strain Rate, s^−1^
0.03–1.05	0–3.1	0–11.5	7.5–35.1	0.05–1	700–1200	10^−4^–20

## Data Availability

Not applicable.

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
