# Peer review of "Prediction of True Stress at Hot Deformation of High Manganese Steel by Artificial Neural Network Modeling"

_materials, 2023, doi:10.3390/ma16031083_

Round 1

Reviewer 1 Report

This paper constructed an artificial neutral network to describe the high-temperature deformation behavior of high Mn steels, which is meaningful for optimizing the hot-deformation performance of the steel. However, there are still some questions must be improved.

1. How the ANN model is built and where is the difference between the ANN model built in this paper as compared to the literatures mentioned about in the introduction? More details about the method and software need to be described specifically.

2. How the concentration, strain, strain rate, and temperature affect the high-temperature flow stress?

3. There is only one experimental data to verify the validity of the model, which is lack of convincing support.

4. The ANN model obtains the effects of Mn content on the effective activation energy, Does Mn content have effects on the other parameter such as A1, A2, A3, n, α, etc.

Author Response

Dear Reviewer!

Thank you for your thorough consideration of our paper “Prediction of True Stress at Hot Deformation of High Manganese Steel by Artificial Neural Network Modeling”, your kind response, and valuable comments. The authors have tried to answer your questions.

COMMENTS FROM REVIEWER

This paper constructed an artificial neural network to describe the high-temperature deformation behavior of high Mn steels, which is meaningful for optimizing the hot-deformation performance of the steel. However, there are still some questions that must be improved.

Reply: the authors have tried to answer the Reviewer's comments and improved the manuscript.

  1. How the ANN model is built and where is the difference between the ANN model built in this paper as compared to the literatures mentioned about in the introduction? More details about the method and software need to be described specifically.

Reply: The ANN-based model was constructed using data from scientific sources (mainly, scientific papers). The main advantage of the constructed model in comparison with the built previously is the following:

Most of the ANN-based models were constructed for one alloy and cannot give information about the influence of the alloying elements on the hot deformation behavior in the Fe-Mn-Al-Si-C system. The current model gives information about the influence of both technological parameters of the thermomechanical treatment (temperature, strain rate, strain) and concentrations of the main alloying elements.

NeuroSolutions 7 software was used for determining the optimal ANN-based model weights values. However, all the calculations by the constructed model were made using our software.

The Materials and Methods part was modified accordingly to the Reviewer’s comment.

  1. How do the concentration, strain, strain rate, and temperature affect the high-temperature flow stress?

Reply: The authors thank the Reviewer for this question. The ANN-based model was constructed especially to answer this question. It is hard to provide all dependencies due to the limited size of the paper. However, the authors try to give the most interesting results obtained up to date (please, see Figures 6-9).

  1. There is only one experimental data to verify the validity of the model, which is lack of convincing support.

Reply: The constructed ANN-based model shows good accuracy for the High-Mn steels in a wide range of the concentration of the alloying elements. It was approved by the testing of the built model on the data from the database that was not used for training (20 % of all records). To approve the workability of the constructed model one more time new 18 compression tests were carried out. Most of them have shown a good convergence with calculated data (please, see Figure 6). However, for each alloy may be obtained only one value of the effective activation energy (for a given true strain value). To approve the workability of our ANN-based model we added a comparison of the Q values for the steels deformed by other scholars (please, see Figure 9a).

  1. The ANN model obtains the effects of Mn content on the effective activation energy, Does Mn content have effects on the other parameter such as A1, A2, A3, n, α, etc.

Reply: Yes, the alloying elements significantly affect all parameters of the constitutive model. The main parameter which describes the hot deformation behavior and lets to compare it with other alloys is the effective activation energy. To prevent the overloading of the manuscript with a large number of graphs we leave just this parameter. However, we have added to Appendix in the revised version the dependencies of the other constitutive model parameters (Please, see Appendix A).

The text of the manuscript was modified accordingly to the Reviewer’s comments.

Reviewer 2 Report

This paper reports the usefulness of the artificial neural network (ANN) based model to predict the flow stress at the high temperature of high manganese austenitic steels with various chemical compositions, interpolatively. Although the model shows good reproductivity of flow stress as shown in Fig.2, the authors use this model only for the confirmation of prediction for the Fe-28Mn-8Al-1C, the relation between the flow stress and Mn or Al content, and the activation energy. Thus, it seems to difficult to find what is the usefulness for the development with the ANN based model. The additional explanation should be necessary with the following points.

(1) Why did the author choose Fe-28Mn-8Al-1C for the experimental materials? This point should be explained very carefully.

(2) Input data.

According to the section 2 and table 1, the input data do not include the grain size which can be changed by the different reheating temperature followed by the different deformation temperature. The reason why the grain size can be neglected should be necessary.

(3) effect of C and Si

Although the effect of the Mn- and Al-bearing is exhibited in Fig.7, there is no information about C and Si. Some additional prediction of the flow stress with these two elements should be presented.

(4) (optional) temperature dependence and activation volume

Activation volume is another index to evaluate the physical meaning of alloying on the flow stress. Is it possible to evaluate the activation volume for Fe-xMn-8Al-1C at various temperature?

Author Response

Dear Reviewer!

Thank you for your thorough consideration of our paper “Prediction of True Stress at Hot Deformation of High Manganese Steel by Artificial Neural Network Modeling”, your kind response, and valuable comments. The authors have tried to answer your questions.

COMMENTS FROM REVIEWER

This paper reports the usefulness of the artificial neural network (ANN) based model to predict the flow stress at the high temperature of high manganese austenitic steels with various chemical compositions, interpolatively. Although the model shows good reproductivity of flow stress as shown in Fig.2, the authors use this model only for the confirmation of prediction for the Fe-28Mn-8Al-1C, the relation between the flow stress and Mn or Al content, and the activation energy. Thus, it seems to difficult to find what is the usefulness for the development with the ANN based model. The additional explanation should be necessary with the following points.

Reply: The constructed ANN-based model shows good accuracy for the High-Mn steels in a wide range of the concentration of the alloying elements (please, see Table 1). It was approved by the testing of the built model on the data from the database that was not used for training (20 % of all records). However, to approve the predictability of the constructed model we made additional 18 experiments with new steel with a composition that differs from the compositions of the steels in the database. It is the usual approach to estimate the accuracy of models.

  1. Why did the author choose Fe-28Mn-8Al-1C for the experimental materials? This point should be explained very carefully.

Reply: We were guided by two main principles during the choice of the “approvement” material composition:

- the composition of the alloy should differ from the steel’s compositions in the database. It is important to approve the reproductivity of the model.

- the composition of the alloy should be near the Fe-30Mn-10Al-1C. This alloy has significant practical applications among lightweight steels. However, even small deviations in the composition (that is a usual situation in industrial production) may significantly influence the hot deformation behavior. The usage of the constructed ANN-based model lets to correct hot deformation technology in dependence on the alloy composition after casting. That provides possibilities to save energy and decrease the material’s losses during production.

  1. Input data. According to the section 2 and table 1, the input data do not include the grain size which can be changed by the different reheating temperature followed by the different deformation temperature. The reason why the grain size can be neglected should be necessary.

Reply: Thank you for your constructive comment. Of course, grain size influences the hot deformation behavior. Unfortunately, it is hard to recognize the change in the grain size during heating before deformation. However, most of the data used for the model construction was obtained by the standard procedures: heating to the high temperature for homogenization, cooling to the deformation temperature, and deformation. The stage of the pre-deformational high-temperature annealing leads to the growth of the grain size to the range of 30 – 100 μm dependent on the steel composition. In our opinion, firstly, such difference hardly has a significant influence on the hot deformation behavior, and, secondly, the grain size before deformation is included in the model through the steel composition. In future work, it will be interesting to include grain size in the model to improve its predictability.

  1. Effect of C and Si. Although the effect of the Mn- and Al-bearing is exhibited in Fig.7, there is no information about C and Si. Some additional prediction of the flow stress with these two elements should be presented.

Reply: We have added to the manuscript the flow stress dependencies on the C and Si content at different temperatures (please, see Figure 8). The text of the manuscript was modified accordingly Reviewer’s comments.

  1. (optional) temperature dependence and activation volume. Activation volume is another index to evaluate the physical meaning of alloying on the flow stress. Is it possible to evaluate the activation volume for Fe-xMn-8Al-1C at various temperature?

Reply: Thank you for your constructive suggestion. We have added the dependence of the activation volume for the Fe-xMn-8Al-1C steel at various temperatures (please, see Figure 9b). The text of the manuscript was modified accordingly Reviewer’s comments.

Reviewer 3 Report

The article presents the simulation of hot deformation of high-manganese steel using Artificial Neural Network Modeling. The validation of the constructed model was carried out by carrying out an additional hot forming test of Fe-28Mn-8Al-1C steel.

The authors recently published an article entitled:

“Churyumov, A.; Kazakova, A.; Churyumova, T. Modeling of the Steel High-Temperature Deformation Behavior Using Artificial Neural Network. Metals 2022, 12, 447. https://doi.org/10.3390/met12030447”

in which they presented an identical deformation simulation for a different steel, coming to very similar conclusions qualitatively.

Additionally, in the article:

“Churyumov, A.Y.; Kazakova, A.A.; Pozdniakov, A.V.; Churyumova, T.A.; Prosviryakov, A.S. Investigation of Hot Deformation Behavior and Microstructure Evolution of Lightweight Fe-35Mn-10Al-1C Steel. Metals 2022, 12, 831. https://doi.org/10.3390/met12050831”

the authors presented studies of hot deformation of Fe-35Mn-10Al-1C steel with almost identical chemical composition to that studied in the submitted manuscript.

I also have doubts about the large number of authorizations (over 18%).

In view of the above, I believe that the article presented to me for review does not bring any scientific novelty. This applies to both the research methodology used and the specific properties of the tested steels.

Therefore, I recommend the article for complete rejection.

Author Response

Dear Reviewer!

Thank you for your consideration of our paper “Prediction of True Stress at Hot Deformation of High Manganese Steel by Artificial Neural Network Modeling”. The authors have tried to answer your questions.

COMMENTS FROM REVIEWER

  1. The authors recently published an article entitled:

“Churyumov, A.; Kazakova, A.; Churyumova, T. Modeling of the Steel High-Temperature Deformation Behavior Using Artificial Neural Network. Metals 2022, 12, 447. https://doi.org/10.3390/met12030447”

in which they presented an identical deformation simulation for a different steel, coming to very similar conclusions qualitatively.

Reply: In our previous paper (https://doi.org/10.3390/met12030447) we developed an approach for the construction of the ANN-based model which included the influence of the chemical composition on the hot deformation behavior. It was applied on the high-alloyed corrosion-resistant steels. In the current work, we developed a new ANN-based model for high-Mn lightweight steels. Despite the similar approaches for the model construction, the scientific novelty of our new paper is not only the new ANN-based model but above all the results which were obtained using the constructed model, such as the influence of the different alloying elements on the flow stress in a wide range of the concentrations and its relation to the phase diagrams of the steels. Besides, the dependences of the main hot deformation characteristics on the Mn concentration in the high-Mn light-weight steels (effective activation energy, activation volume) were obtained using constructed model.

  1. Additionally, in the article:

“Churyumov, A.Y.; Kazakova, A.A.; Pozdniakov, A.V.; Churyumova, T.A.; Prosviryakov, A.S. Investigation of Hot Deformation Behavior and Microstructure Evolution of Lightweight Fe-35Mn-10Al-1C Steel. Metals 2022, 12, 831. https://doi.org/10.3390/met12050831”

the authors presented studies of hot deformation of Fe-35Mn-10Al-1C steel with almost identical chemical composition to that studied in the submitted manuscript.

Reply: The compositions of the investigated steels in both papers are far to be identical. Mn and Al have a significant influence on the stacking fault energy of the austenite phase. As a result, any changes in the concentrations of these elements influence the hot deformation behavior. The difference in the Mn concentration is 7 %, and the difference in Al content is 2 %. The overall change in the composition of 9 % made significant changes in the hot deformation behavior of the steels (please, compare the effective activation energies for both steels).

  1. I also have doubts about the large number of authorizations (over 18%).

Reply: The authors carefully rewrite the manuscript to prevent a large per cent of the authorization.

Round 2

Reviewer 2 Report

On the reply about the previous comment 4, please check the unit (m^3?) of Fig.9(b).

(The peak appearing around 26%Mn is inetesting. The activation volume can be related to the product of Burgers vector and the spacing of the pinning points. These should be changed by both SFE and the atomic fraction of solutes, and these are controlled by Mn content, although it appears difficult to discuss these relations precisely as the authors mention in the added text, 'The nature of this phenomenon requires detailed study'(P.9, L.233).  The reviewer hopes additional future success of the authors studies.)

Author Response

Dear Reviewer!

Thank you for your kind response to the revised version of our manuscript “Prediction of True Stress at Hot Deformation of High Manganese Steel by Artificial Neural Network Modeling. The authors have corrected the units in Figure 9b. In our future investigations, we try to recognize the reason for the peak in the dependence of the activation volume on the Mn content.

Reviewer 3 Report

The authors responded to my comments.

I believe that the novelty presented in the article is negligible. The content of the article does not significantly extend the existing knowledge in the field of research methodology as well as in the field of testing materials and their properties. The authors supplemented References by one item. This does not change my assessment that the rate of self-citation remains very high.

I believe that the level of novelty is insufficient to be published in such a reputable journal with a high Impact Factor as Materials.

Author Response

Dear Reviewer!

Thank you for your consideration of our paper “Prediction of True Stress at Hot Deformation of High Manganese Steel by Artificial Neural Network Modeling”. The authors have tried to answer your question.

COMMENTS FROM REVIEWER

  1. I believe that the novelty presented in the article is negligible. The content of the article does not significantly extend the existing knowledge in the field of research methodology as well as in the field of testing materials and their properties. The authors supplemented References by one item. This does not change my assessment that the rate of self-citation remains very high. I believe that the level of novelty is insufficient to be published in such a reputable journal with a high Impact Factor as Materials.

In the current work, we developed a new ANN-based model for high-Mn lightweight steels. The scientific novelty of our new paper is not only the new ANN-based model but above all the results which were obtained using the constructed model, such as the influence of the different alloying elements on the flow stress in a wide range of concentrations and its relation to the phase diagrams of the steels. Besides, the dependences of the main hot deformation characteristics on the Mn concentration in the high-Mn light-weight steels (effective activation energy, activation volume) were obtained using constructed model. For example, interesting non-monotonic dependence of the activation volume on the Mn content in the range of 20 – 30 % was found. The nature of this phenomenon requires future detailed study.

The authors have analyzed more works about the application of artificial neural networks in the materials science field and decreased the self-citation index from 18 to 15 %.
